# Melanocyte Activation Mechanisms and Rational Therapeutic Treatments of Solar Lentigos

**DOI:** 10.3390/ijms20153666

**Published:** 2019-07-26

**Authors:** Genji Imokawa

**Affiliations:** Center for Bioscience Research & Education, Utsunomiya University, 350 Mine Utsunomiya, Tochigi 321-8505, Japan; imokawag@dream.ocn.ne.jp; Tel.: +81-28-649-5282

**Keywords:** solar lentigo, endothelin, stem cell factor, keratinocyte growth factor, interleukin-1, tumor necrosis factor α, intracellular signaling, calcium mobilization, signaling blocker, *M. chamomilla*, ascorbate-phosphate Na, whitening agent, tyrosinase inhibitor

## Abstract

To characterize the pathobiology of solar lentigos (SLs), analyses by semiquantitative RT-PCR, Western blotting, and immunohistochemistry revealed the upregulated expression of endothelin (EDN)-1/endothelin B receptors (EDNBRs), stem cell factor (SCF)/c-KIT, and tumor necrosis factor (TNF)α in the lesional epidermis, which contrasted with the downregulated expression of interleukin (IL) 1α. These findings strongly support the hypothesis that previous repeated UVB exposure triggers keratinocytes to continuously produce TNFα. TNFα then stimulates the secretion of EDNs and the production of SCF in an autocrine fashion, leading to the continuous melanogenic activation of neighboring melanocytes, which causes SLs. A clinical study of 36 patients with SLs for six months treated with an *M. Chamomilla* extract with a potent ability to abrogate the EDN1-induced increase in DNA synthesis and melanization of human melanocytes in culture revealed a significant improvement in pigment scores and color differences expressed as L values. Another clinical study using a tyrosinase inhibitor L-ascorbate-2-phosphate 3 Na (ASP) demonstrated that L values of test lotion (6% APS)-treated skin significantly increased in SLs and in non-lesional skin with a significantly higher ΔL value in SLs when compared with non-lesional skin. The sum of these findings strongly suggests that combined topical treatment with EDN signaling blockers and tyrosinase inhibitors is a desirable therapeutic choice for SLs.

## 1. Introduction

The most frequently appearing epidermal hyperpigmentary disorder in Asian skin is solar lentigos (SLs), which generally occur on the face and on the dorsum of the hands. In contrast to UVB-induced hyperpigmentation (UVB melanosis), which, depending on the age of the subject, disappears within two weeks to a few months after discontinuation of the UVB exposure, SLs develop on sun-exposed skin, especially the face, and never disappear due to possible DNA damage in the keratinocytes elicited by repeated UVB irradiation of the lesional epidermis. In general, hyperpigmentary disorders are generally targeted by anti-pigmenting agents and include UVB-melanosis, SLs, and melasma. Based on the frequency of the final diagnosis for patients with various pigmentary disorders in Japan, SLs have the highest incidence, occurring in approximately 60% of all patients with hyperpigmentary disorders, while melasma and post-inflammatory hyperpigmentation (including UVB-melanosis) occur in as low as 5.2% and 3.3% of patients, respectively [1]. This suggests that SLs are a predominant target of anti-pigmenting agents for Asian skin. However, there have been few published papers on the clinical effects of anti-pigmenting agents on SLs because many anti-pigmenting agents serve as tyrosinase inhibitors and it is not anticipated that tyrosinase inhibition would be sufficient to ameliorate the hyperpigmentation of SLs. Additionally, it seems difficult to rationally design an effective therapeutic topical treatment for SLs because little is known about the melanocyte activation mechanisms in the lesional epidermis. In this review article, we characterize the pathobiology of SLs according to the known melanogenic paracrine cytokine networks, and based upon the discovered melanocyte activation mechanisms in the lesional epidermis of SLs, we introduce rational clinical approaches for the topical treatment for SLs to ameliorate the hyperpigmentation level.

## 2. Clinical Characteristics

SLs clinically develop as flat, well-circumscribed patches of skin with varying colors and sizes, frequently appearing on the face and on the dorsum of the hands (Figure 1a) [2]. The histochemistry of SL lesions stained with Hemoxylin & Eosin revealed slight acanthosis and pigmentation along the basal cell layer (Figure 1b) [2]. There are two patterns in terms of the histopathological features of SLs on the face: one pattern demonstrated a flattened epidermis with basal melanosis, and the other pattern showed epidermal hyperplasia with elongated rete ridges composed of deeply pigmented basaloid cells [3].

## 3. Mechanisms of Melanocyte Activation in Solar Lentigo Based upon Melanogenic Paracrine Cytokine Network

### 3.1. Melanocyte Number and Tyrosinase Expression

Although there have been some arguments on the increased number of melanocytes in SLs, immunohistochemical analysis using the melanocyte-specific marker MART-1 revealed an increased number of melanocytes in solar lentigo [5,6,7]. However, the actual density of the melanocytes along the border between the dermis and epidermis in SLs is similar to that in perilesional control skin due to the increased proliferation of keratinocytes [7]. Immunohistochemistry using anti-tyrosinase revealed that tyrosinase-positive melanocytes were significantly increased by 2-fold in the lesional epidermis of SLs [4]. Gene analysis by semiquantitative RT-PCR revealed that the tyrosinase mRNA expression level is significantly upregulated by 2.3-fold in the lesional epidermis [4]. The above findings support the possibility that there is a stimulation of both proliferation and melanization in SL lesional melanocytes. Therefore, we hypothesized that slightly proliferating keratinocytes in SLs trigger the activation of neighboring melanocytes by secreting melanocyte-stimulating cytokines.

### 3.2. Melanogenic Paracrine Cytokine Networks

We, and other groups, have elucidated that there are several important melanogenic paracrine cytokine networks between skin cells (Figure 2). Mainly, these include endothelin (EDN)-1 [8,9,10,11,12,13,14,15], membrane-bound stem cell factor (mSCF) [16], proopiomelanocortin (POMC) [17,18,19,20], prostaglandin E_2_ [21], granulocyte macrophage colony stimulation factor (GM-CSF) [22], basic fibroblast growth factor [23], growth-related oncogene α [24] and keratinocyte growth factor (KGF) [25,26] involved in keratinocyte/melanocyte interactions and soluble SCF [27], hepatocyte growth factor (HGF) [8,10,27,28,29] and KGF [30] involved in fibroblast/melanocyte interactions. Based upon the elucidated melanogenic paracrine cytokine networks including their corresponding receptors, it is important to determine which melanogenic paracrine cytokine networks are involved and are specifically activated in vivo in the hyperpigmentation mechanisms in the lesional epidermis of SLs.

### 3.3. Major Paracrine Cytokines and Receptors Responsible for Melanocyte Activation in SLs

Among the above melanogenic cytokines and their corresponding receptors, we first determined the role of basic fibroblast growth factor (bFGF) and growth-related oncogene α (GROα) in the epidermis of SLs. Basic FGF was found to be overexpressed in UVB-exposed human keratinocytes, whose homogenates had the distinct potential to stimulate melanogenesis and the proliferation of human melanocytes in culture, although cofactors with the ability to increase cyclic AMP levels are essentially required for melanogenic stimulation [23]. GROα was identified by our research group as a melanogenic cytokine that plays an essential role in phenylazo-naphtol-induced hyperpigmentation after its allergic reaction in brownish yellow guinea pig skin [24,31]. Semiquantitative RT-PCR revealed that there was no change in the gene expression levels of bFGF and GROα in the SL lesional epidermis when compared with non-lesional epidermis [2]. Consistent with the mRNA expression level of bFGF and GROα, immunohistochemistry revealed that there was no difference in the immunostaining intensity with anti-bFGF (Figure 1c,d) and anti-GROα (Figure 1e,f) between the SL lesional and non-lesional epidermis [2]. These findings indicated no involvement of bFGF or GROα as intrinsic melanogenic cytokines for the melanocyte activation mechanism in SLs.

In the biological mechanism of UVB-induced hyperpigmentation, the expression of EDN1, a vasoconstrictor peptide originally isolated from porcine endothelial cells [32], and mSCF [16] are upregulated in an autocrine fashion by the action of the UVB-stimulated release of interleukin (IL)-1α via the generation of reactive oxygen species (ROS) [15]. Those cytokines cause neighboring melanocytes to increase their expression of the critical melanin synthesizing enzymes tyrosinase [14,33], EDNBR [34], and melanosome matrix protein PMEL17 [34] as well as proliferation-related enzymes such as cyclic dependent kinase (CDK)2 [34], which lead to hyperpigmentation in UVB-exposed skin. Based upon the major melanogenic cytokines essentially involved in UVB-melanosis, we next determined the role of EDN1 in the melanocyte activation mechanism in the epidermis of SLs. Characterization of the mRNA expression level of EDN1 in the epidermis by semiquantitative RT-PCR analysis showed that there was a marked increase in expression by an average of 3.2-fold in the SL lesional epidermis when compared with non-lesional epidermis [4]. Consistent with the increased gene expression level of EDN1, there was increased immunostaining with anti-EDN1 throughout the SL lesional epidermis (Figure 1g,h) [4].

In addition to EDN1, we next determined the role of EDN receptors in the melanocyte activation mechanism underlying SLs. The binding of EDN to its receptor is the first step in the major paracrine lineage between keratinocytes and melanocytes that upregulates skin pigmentation [11,14,35,36]. EDN receptors are seven-transmembrane G-protein coupled receptors with two isoforms (A and B) that interact specifically with EDN1 and with all forms of EDNs (EDN1, EDN2, and EDN3), respectively [37]. Regarding the inhibitory effects of EDN receptor antagonists on the EDN-stimulated proliferation of human melanocytes in culture, a significant inhibitory effect occurred only in the presence of BQ 788, an endothelin B receptor (EDNBR) antagonist, but not of BQ123 or BQ610, an endothelin A receptor (EDNAR) antagonist [9], which indicated that EDN1/EDN receptor signaling is mediated via EDNBR. Semiquantitative RT-PCR analysis of the expression of EDNBR mRNA revealed that among the various types of skin cells, melanocytes are the only substantial type of cell expressing EDNBR. Since we have already demonstrated that EDN1 secreted by keratinocytes triggers the activation of the intracellular protein kinase C (PKC) via EDNBR [35], we determined whether the expression level of EDNBR was also accentuated in melanocytes in the lesional SL epidermis. Semiquantitative RT-PCR analysis of EDNBR mRNA in the epidermis of SLs demonstrated a markedly increased expression by an average 6.8-fold in the lesional SL epidermis [4]. Consistent with the increased expression of EDNBR mRNA, there was increased immunostaining with anti-EDNBR localized in melanocytes in the lesional SL epidermis (Figure 1i,j) [4]. The sum of these findings strongly suggest the coordinated increase in the expression of EDN1 and its receptor linkage in the lesional epidermis of SLs. 

Based upon the major melanogenic cytokines involved in UVB-melanosis, we next determined the role of SCF in the melanocyte activation mechanism in the lesional SL epidermis. We have already reported that UVB exposure of cultured human keratinocytes as well as human epidermis significantly stimulates the expression of SCF at both the gene and protein levels [15,16]. Semiquantitative RT-PCR analysis of SCF mRNA in the epidermis of SLs demonstrated the markedly increased expression by an average of 3.9-fold in the lesional SL epidermis compared with non-lesional epidermis [2]. Western blotting for SCF in the skin of six patients with SLs demonstrated that there was a significant increase by an average of 1.6-fold in SCF protein in the lesional SL epidermis compared with non-lesional epidermis [2].

Consistent with the increased gene and protein expression level of SCF, there was increased immunostaining with anti-SCF throughout the lesional SL epidermis (Figure 1k,l) [2]. There was an argument about whether the SCF upregulated at the protein level in the lesional epidermis is the soluble type or the membrane-bound type. If basement membrane-permeable-soluble SCF is upregulated in the epidermis, mast cells present in the dermis should be activated to proliferate and increase in number. Since toluidine blue staining in SLs did not show any increase in the number of mast cells in the lesional SL dermis [2], it is likely that the type of SCF upregulated in SLs is the membrane-bound type. 

In addition to SCF, we next determined the role of the SCF receptor c-KIT in the melanocyte activation mechanism underlying SLs. We have already reported that UVB exposure of cultured human melanocytes as well as human epidermis significantly stimulates the expression of c-KIT at both the gene and protein levels [15,38]. Furthermore, in the UVB-stimulated pigmentation of brownish yellow guinea pig skin, a blocking antibody to c-KIT significantly abrogated the increased number of dopa-positive melanocytes as well as hyperpigmentation in an early phase of the UVB-induced pigmenting process [16], which strongly indicates that the binding of SCF to c-KIT plays an important role in the UVB-induced activation of melanocytes, leading to hyperpigmentation. Concomitant with the increased expression of SCF at the gene and protein levels, the gene expression level of its c-KIT receptor was also increased by an average 2.14-fold in the lesional SL epidermis [2]. Consistent with the increased expression of c-KIT mRNA, there was increased immunostaining with the c-KIT antibody localized in melanocytes in the lesional SL epidermis (Figure 1m,n) [2]. This suggests that there is a coordinated increase in the expression of SCF and its receptor c-KIT in the lesional SL epidermis.

On the other hand, other groups have hypothesized that keratinocyte growth factor (KGF)/KGF receptor (KGFR) play an important role in the initiation of SL formation and in increased pigmentation only in the epidermis of earlier SLs stages [25]. In the lesional epidermis [30] and/or dermis [25] of SLs, KGF expression is upregulated only at the immune-staining level although further studies on its gene and protein expression are required for any final conclusions on it as an intrinsic cytokine causing SLs. While IL-1α is known to cause keratinocytes to stimulate the production of KGF, it remains unclear as to how KGF expression is upregulated in the lesional epidermis where IL-1α expression is rather downregulated [2]. Of course, it should be determined as to whether TNFα (that is upregulated in the lesional epidermis) has an ability to stimulate KGF production. Since KGF expression in the skin is primarily confined to dermal fibroblasts, it is plausible that the increased expression of KGF by dermal fibroblasts in the lesional dermis of SLs [25] is associated with both the increased proliferation of keratinocytes and the stimulated melanogenesis. Thus, it is likely that dermal fibroblasts are persistently activated by UV exposure to release KGF, which acts directly or indirectly thorough keratinocytes to modulate the expression of SCF, contributing to the hyperpigmentation of SL [25]. 

We next determined the biological mechanisms by which EDN1 secretion is upregulated in the lesional epidermis of SLs. It is well known that biological factors associated with the stimulation of EDN secretion by human keratinocytes include IL-1α, tumor necrosis factor (TNF) α, and endothelin converting enzyme (ECE)-1α. Our findings on the autocrine cytokine stimulation associated with UVB melanosis demonstrated that the upregulation of IL-1α is mainly responsible for stimulating the production of EDN1 and SCF in UVB-exposed human keratinocytes [12,16]. Thus, IL-1 α and β have been found to be potent stimulators of EDN secretion with a delayed peak in human keratinocytes in culture that resembles the pattern of the UVB-induced secretion of EDN [11]. It is also well known that UVB irradiation significantly stimulates the release of IL-1α but not IL-1β in cultured human keratinocytes and that a blocking antibody to IL-1α significantly abrogates the increased secretion of EDN1 [11], which indicates that UVB-induced hyperpigmentation is mediated via activation of epidermal melanocytes resulting from an increased secretion of EDN1 by UVB-exposed keratinocytes. Additionally, the expression of mSCF protein has been found to be significantly stimulated by treatment with IL-1α in human keratinocytes in culture [16]. Therefore, we next determined whether IL-1α was also upregulated to stimulate the expression of EDN1 and/or SCF in the lesional epidermis of SLs. Interestingly, semiquantitative RT-PCR analysis revealed that the expression of IL-1α mRNA is rather down-regulated in the lesional epidermis [16]. Consistent with that RT-PCR analysis, immunohistochemistry with an IL-1α antibody revealed a weaker immuno-staining in the lesional epidermis than in the non-lesional epidermis (Figure 1o,p) [2], which suggested that IL-1α is not responsible for the increased expression of EDN1 and SCF in the lesional epidermis of SLs.

Regarding mechanisms underlying the increased expression of EDN1 in SLs, in addition to IL-1α, TNFα at 10 ng/mL was reported by Tsuboi et al. to be a potent stimulator (by 10-fold) of EDN secretion by cultured human keratinocytes [39]. As for the mechanism underlying the increased expression of SCF in SLs, we next examined the effects of TNFα on SCF expression in cultured human keratinocytes. Western blotting using an anti-SCF antibody demonstrated that TNFα significantly stimulates the production of mSCF [15]. Therefore, we next determined whether or not TNFα is upregulated to stimulate the expression of EDN1 and/or SCF in the lesional epidermis of SLs. Interestingly, semiquantitative RT-PCR analysis revealed that the expression of TNFα mRNA is significantly upregulated in the lesional SL epidermis [2]. Consistent with the RT-PCR analysis, immunohistochemistry with an anti-TNFα antibody revealed stronger immunostaining in the lesional SL epidermis than in the non-lesional epidermis (Figure 1q,r) [2]. Thus, it was conceivable that the upregulated expression of TNFα is mainly responsible for the increased expression of EDN1 and SCF in the lesional epidermis of SLs. 

As for the mechanism involved in the increased expression of EDN1, we next determined whether ECE-1α is upregulated in the lesional SL epidermis. As no study has described the expression of ECE-1α in human keratinocytes at this time, we characterized ECE-1α in cultured human keratinocytes compared with endothelial cells. Analysis of ECE-1α activity in various types of skin cells revealed that human keratinocytes and not human melanocytes possess ECE-1α activity, which occurs to a lesser extent than in endothelial cells [40]. Western blotting, using antibodies we prepared to ECE-1α, demonstrated that the ECE-1α protein exists in human endothelial cells, human fibroblasts, and human keratinocytes, but not in human melanocytes [40]. Assays of ECE-1α activity in supernatants following immunoprecipitation with the ECE-1α antibody demonstrated that endothelial cells and human keratinocytes have detectable activities of ECE-1α at pH 6.8, which correlates well with the ECE-1α immunoprecipitated with our antibody to ECE-1α [40]. Semiquantitative RT-PCR analysis of ECE-1α mRNA levels revealed that IL-1α, but not TNFα, had a slight stimulatory effect on ECE-1α gene expression in cultured human keratinocytes [40]. Consistent with the RT-PCR analysis, Western blotting revealed that IL-1α, but not TNFα, stimulated ECE-1α protein expression in cultured human keratinocytes [40]. Finally, we determined whether ECE-1α is also upregulated to stimulate EDN1 expression in the lesional epidermis of SLs. Consistent with the lack of a stimulatory effect of TNFα, which is upregulated in SLs, there was no difference in the mRNA expression level of ECE-1α between the lesional and the non-lesional epidermis of SLs [2], which suggests that ECE-1α is not responsible for the increased secretion of EDN1 in SLs. 

Figure 3 shows a summary of autocrine stimulation in human keratinocytes in culture. As for the biological mechanisms leading to the activation of EDN and SCF signaling cascades [9], our in vitro studies suggest an interesting contrast that while the upregulation of IL-1α is mainly responsible for stimulating EDN1 and SCF production in UVB-melanosis, the upregulation of TNFα is mainly associated with the stimulated production of those same two cytokines in SLs. 

### 3.4. Synergistic Stimulatory Effects of the Combination of EDN1 and SCF

We have already reported that there is a synergy in the stimulated DNA synthesis in cultured human melanocytes as measured by ^14^C-thymidine incorporation in the co-presence of SCF and EDN1 [41]. There was also a similar synergy in the stimulation of melanin synthesis as measured by ^14^C-thiouracil incorporation in cultured human melanocytes in the co-presence of SCF and EDN1 [41]. In contrast, there was no such synergy between EDN1 and the granulocyte macrophage colony stimulatory factor (GM-CSF) or HGF [13,42]. As for the signaling mechanisms involved in those synergistic effects, we found that the cross-talk between SCF and EDN1 signaling was initiated through the tyrosine phosphorylation of c-KIT that is stimulated indirectly by activated PKC, which enhances the formation of the Shc-Grb2-SOS complex that leads in turn to the synergistic activation of the Ras/Raf-1/MEK/MAP kinase loop [41].

### 3.5. Mutual Interactions between EDN/EDNBR and SCF/c-KIT

We next determined if the increased production of SCF triggers EDNBR expression or its affinity to the EDN ligand in human melanocytes in addition to its stimulatory effect on their proliferation. Western blotting analysis revealed that SCF can stimulate the expression of EDNBR protein in cultured human melanocytes [43]. When the affinity of EDNBR to its ligand was evaluated in cultured human melanocytes using a ligand binding assay, the binding of ^125^I-lableled EDN1 to EDNBR was found to be significantly increased two days after incubation with SCF [43]. Taken together, these findings indicate that SCF expressed in the early phase may enhance the expression of EDNBR, which cause melanocytes to become more sensitive to the later secretion of EDN1. On the other hand, when cultured human melanocytes were treated for 48 h with EDN1 at 10 nM, the ligand binding assay using ^125^I-SCF revealed that EDN1 distinctly enhanced the binding affinity of SCF to the c-KIT receptor [43].

## 4. Summary of the Pathobiology of SLs

Table 1 shows a summary of the paracrine cytokine networks that occur in various epidermal hyperpigmentary disorders. With respect to those networks, SLs are very similar to UVB-melanosis except for the causative cytokines such as TNFα for the increased production of EDN1 and SCF.

As for the biological mechanisms leading to the increased activities of the EDN and SCF signaling cascades, our in vitro studies suggested that while the upregulation of IL-1α is mainly responsible for stimulating EDN1 and SCF production in UVB-melanosis, the upregulation of TNFα is associated with the stimulated production of those same two cytokines in SLs. Figure 4 shows a summary of the complex relationships between the SCF and EDN1 linkages in the epidermis of SLs. That includes synergism between SCF and EDN1 as well as the activation of EDNBR and c-KIT by SCF and by EDN1, respectively. The release of TNFα by keratinocytes simulates the production of SCF and EDN in an autocrine fashion, both of which exhibit a synergistic effect on melanocyte activation as well as a stimulatory effect on the expression of their corresponding receptors, c-KIT and EDNBR. These synergistic and intercellular interactions facilitate melanocyte activation to a larger extent in the lesional epidermis of SLs than in the UVB-exposed epidermis, leading to the more intensive hyperpigmentation of SLs.

Collectively, our results suggest that two signaling cascades, EDN1/EDNBR and mSCF/c-KIT, play an intrinsic and coordinated role in accentuating the mitogenesis and melanogenesis of melanocytes in the hyperpigmented epidermis of SLs. Figure 5 shows the biological sequence for hyperpigmentation mechanisms involved in SLs, where unknown tumorigenic factors due to cumulative DNA damage in the distant past cause keratinocytes to produce and secrete TNFα. Thus, TNFα causes keratinocytes to overproduce melanogenic cytokines such as SCF and EDN1 in an autocrine fashion, triggering adjacent melanocytes to stimulate melanin synthesis, leading to epidermal hyperpigmentation.

## 5. Therapeutic Topical Treatment Approaches

Based on the coordinated melanogenic paracrine network and activated signaling mechanisms leading to melanocyte activation in the lesional SL epidermis, blocking essential melanogenic intracellular signaling is a desirable therapeutic approach to achieve anti-pigmenting effects on SLs. Such an approach might cause hypopigmentation, but should not be effective in non-lesional skin where such signaling cascades are not activated in melanocytes. On the other hand, inhibiting tyrosinase activity is another approach to ameliorate the hyperpigmentation in SLs, although it might cause hypopigmentation and might also be effective in non-lesional skin.

### 5.1. Blocking Essential Melanogenic Intracellular Signaling

Since almost all mutations leading to genetic hypopigmentary disorders occur in the EDN1/EDNBR and SCF/c-KIT axis [44] and since there is a synergistic stimulatory effect of EDN1/SCF on cell proliferation and melanization in cultured human melanocytes, it is conceivable that blocking either EDN1/EDNBR or SCF/c-KIT signaling could prevent hyperpigmentation due to the coordinately increased expression of EDN1 and SCF because a synergistic stimulatory effect fails to occur. Therefore, we attempted to achieve anti-pigmenting effects on SLs by blocking the EDN/EDNBR signaling lineage.

The EDN-activated intracellular signaling pathway consists of binding to EDNBR, activating PKC, the MAP kinase cascade, and the cAMP/PKA cascade [34,35,41]. Thus, these cellular actions are initiated by the binding of EDN1 to the G-protein-coupled EDNBR, followed by sequential signaling processes consisting mainly of PKC and MAPK. After binding to its receptor, EDN1 triggers the hydrolysis of polyphosphoinositide by activating phospholipase C, which generates inositol-trisphosphates (IP3) and diacylglycerol, mobilizing intracellular Ca^++^ and activating PKC, respectively. PKC activation is attained by its translocation from the cytosol to the plasma membrane and activates Raf-1 through phosphorylation. Thus, Raf-1 appears to be a convergent point between the PKC and MAPK pathways. Raf-1 activation leads to the activation of a series of MAPK pathway intermediates consisting of MEK, ERK, and RSK. Phosphorylated Raf-1 activates MEK by phosphorylation and activated MEK phosphorylates ERK. The activated ERK then phosphorylates microphthalmia-associated transcription factor (MITF) at serine 75, leading to the recruitment of a co-activator for regulating the gene expression of several melanogenic factors [41]. Simultaneously, activated MAPK results in the activation of RSK, which phosphorylates CREB, leading to the transcription of MITF. On the other hand, activated PKC has cross-talk with the adenyl cyclase cascade to produce cAMP [8], leading to the activation of PKA, which also activates CREB by phosphorylation, leading to the increased expression of MITF. The increased level of MITF protein stimulates the expression of melanocyte-specific genes including tyrosinase, PMEL17, EDNBR, c-KIT, and CDK2.

Since several intracellular signaling pathways lead to stimulated melanogenesis within melanocytes, and the EDN signaling cascade is specifically associated with the PKC pathway, which involves calcium mobilization from the endoplasmic reticulum [13], we used a calcium mobilization assay to screen for EDNBR antagonists from a variety of herbal extracts. Calcium mobilization from the endoplasmic reticulum occurs after the generation of IP_3_ and diacylglycerol due to the hydrolysis of polyphosphoinositide through activated phospholipase Cγ. When human melanocytes are treated in culture with EDN1, the mobilization of calcium detectable by the fura-2AM reagent produces fluorescence after binding to released calcium, as seen by the rapid appearance of a yellow color that can be measured in real time by digital imaging microscopy. From screening many herbal extracts, we found that pre-incubation with a Matricaria *chamomilla* extract interrupted the calcium mobilization induced by EDN1 [1], which suggested that it can serve as an effective antagonist against EDNBR. *M. chamomilla*, commonly known as chamomile, is an annual plant of the composite family *Asteraceae*. *M. chamomilla* is the most popular source of the herbal product chamomile, although other species are also used as chamomile. 

Based upon the inhibitory effect of samples fractionated from the *M. chamomilla* extract on EDN1-induced calcium mobilization in cultured human melanocytes, we identified spiroether as its active compound with a potent ability to interrupt calcium mobilization (Figure 6) [1]. There are two isomers of spiroether (E and Z), and the spiroether E-isomer can completely abolish calcium mobilization at concentrations of more than 1 µM. Treatment with the spiroether E-isomer at 0.2 and 1.0% concentrations significantly reduced the UVB-induced pigmentation in brownish yellow guinea pig skin as evaluated by the ΔL value, which indicated that blocking the EDN-mediated signaling cascade is effective in preventing UVB-induced hyperpigmentation. This is important in vivo evidence that shows the intrinsic involvement of the EDN cascade in UVB-melanosis. On the other hand, as expected, when pre- or co-incubated in melanocyte culture, the *M. chamomilla* extract had a potent ability to abolish the EDN1-induced increase in DNA synthesis (as measured by ^14^C-thymidine incorporation) as well as melanin synthesis (as measured by ^14^C-thiouracil incorporation) by cultured human melanocytes. In a parallel study using cultured human keratinocytes, we verified that the *M. chamomilla* extract had no inhibitory effect on the IL-1α-induced secretion of EDN1 [1]. Additionally, using tyrosinases derived from human melanocytes, we confirmed that the *M. chamomilla* extract had no inhibitory effect on tyrosinase activity in vitro in contrast to the distinct inhibitory effects by the well-known whitening agents, kojic acid and arbutin [1]. When the *M. chamomilla* extract was topically applied daily to brownish yellow guinea pig skin for two weeks immediately after UVB irradiation, the intensity of the UVB-induced pigmentation was significantly decreased when compared with treatment with 10% arbutin or vehicle only [1].

In a human clinical study, the topical application of the *M. chamomilla* extract to UVB-exposed human forearm skin for six weeks immediately after the irradiation was found to significantly prevent UVB-induced hyperpigmentation as measured by a color difference meter and expressed as a ΔL value. 

Using an *M. chamomilla* extract-containing stick type wax, we examined the clinical effects on pigmentation in SLs. In this clinical study, each pigmented spot was examined for color changes and the degrees of clinical improvement and validity were evaluated. A clinical evaluation performed at the Tokyo Women’s Medical University revealed that treatment for two months with the *M. chamomilla* extract resulted in 48% of subjects showing a marked improvement and 40% with slight improvement [1]. In the changes of pigmentation level measured by ΔL values, all patients with SLs had distinct increases over two in their ΔL value, a visibly recognizable level, after treatment for 2~3 months. There was a significant decrease in the pigmentation level as measured by ΔL values between 0, 1, and 2 months after treatment (Figure 7). Overall, the clinical evaluation for three months revealed that treatment with the *M. chamomilla* extract resulted in 42% of subjects showing a marked improvement, 12% with a moderate improvement, and 25% with slight improvement. Another clinical trial performed in two related dermatological hospitals in Tokyo demonstrated that treatment with the *M. chamomilla* extract gradually reduced the pigmentation and after six months, resulted in approximately 70% of subjects showing more than slight improvement including 10% with disappearance and 15% with marked improvement. There were two cases where the SLs completely disappeared after approximately six months of treatment, with ΔL values increasing up to 8.4 and 6.9 after six months of treatment (Figure 8) [1].

In conclusion, regarding therapeutic approaches for SLs by blocking essential melanogenic intracellular signaling, the above clinical studies suggest that blocking EDN1/EDNBR-associated signaling is an effective therapeutic treatment for SLs without any hypopigmenting effects.

### 5.2. Inhibiting Tyrosinase Activity

Inhibiting tyrosinase activity is another approach for ameliorating the hyperpigmentation in SLs that has possible drawbacks of hypopigmentation, but the potential benefit of also being effective in non-lesional skin. As whitening agents are generally designed to treat UVB-melanosis, there has not been a clinical double-blinded study for the potential anti-pigmenting effects of whitening agents on SLs. We conducted a double-blinded half face study on 27 Japanese female volunteers with SLs. In that clinical study, lotions with or without 6% L-ascorbate-2-phosphate 3 Na (APS) (test lotion/placebo lotion, respectively) were topically applied twice daily on the face for 24 weeks [45]. Representative photographs of the faces of subjects No. 012 and No. 016 before and after treatment demonstrated that the pigmentation level of the test lotion-treated SLs appeared to slightly decrease in concert with a slight decrease in the test lotion-treated non-lesional skin [45]. In contrast, the pigmentation level of the placebo lotion-treated SLs and the non-lesional skin appeared to remain unchanged. Whereas L values in the test lotion-treated SLs significantly increased after 24 weeks of treatment, L values in the placebo lotion-treated SLs remained unchanged (Figure 9) [45]. Comparisons of △L values before and after the treatments revealed a significantly higher △L value in the test lotion-treated SLs than in the placebo lotion-treated SLs (Figure 9). These findings suggest that the APS-containing test lotion had a significantly stronger anti-pigmenting effect on SLs than the placebo lotion. Whereas L values in the test lotion-treated non-lesional skin significantly increased after the treatment for 24 weeks, L values in the placebo lotion-treated non-lesional skin remained unchanged. Comparison of the △L values before and after the treatments revealed a significantly higher △L value in the test lotion-treated non-lesional skin than in the placebo lotion-treated non-lesional skin [45]. These findings suggest that the test lotion had a significantly stronger whitening effect on non-lesional skin than the placebo lotion. Comparison of the anti-pigmenting effects between SLs and non-lesional skin revealed that while there were significantly higher △L values between the before and after treatment with the test lotion in SLs than in the non-lesional skin, the △L values occurred at a similar level in the SLs and non-lesional skin without any significant difference between them [45]. 

The relationship between the L values and melanin indices for all measurements during this clinical study revealed a significant correlation between them both. This result indicated that a 2.0 ΔL value was approximately equivalent to a 20 Δ melanin index with a 10-fold higher sensitivity in the melanin index than the L value. A comparison between the L values and melanin indices for each measurement at weeks 0 and 24 on SLs revealed significant correlations. These comparisons between the L values and melanin indices indicated that there was a marked distribution shift toward a lighter color such as a greater L value and a smaller melanin index in the test lotion-treated SLs whereas there was no such distinct distribution shift in the placebo lotion-treated SLs [45].

In conclusion, regarding a therapeutic approach for SLs using a tyrosinase inhibitor, the sum of the above findings strongly indicates that APS has a weak but significant anti-pigmenting effect on SLs and a significant whitening effect even on normal colored healthy skin, without any hypopigmenting effect.

## 6. Conclusions

In conclusion, regarding the melanocyte activation mechanisms and therapeutic topical treatment of SLs, based on the above findings of the melanocyte activation mechanism in SLs as well as the clinical efficacy obtained using an EDN signaling blocker and a tyrosinase inhibitor, it is strongly suggested that combined treatment of an EDN signaling blocker and a tyrosinase inhibitor is a desirable therapeutic treatment for SLs.

## Figures and Tables

**Figure 1 ijms-20-03666-f001:**
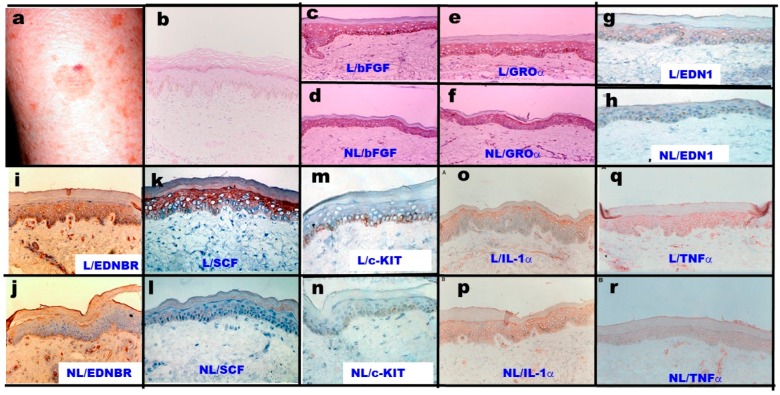
Clinical appearance of SLs and the immunochemistry of SLs and non-lesional skin with antibodies to various melanogenic cytokines and receptors [2,4]. L: Lesion, NL: Non-lesion. (**a**) Clinical appearance of SLs on the forearm; (**b**) Hemoxylin & Eosin staining; (**c**,**d**) Immunohistochemistry with anti-bFGF; (**e**,**f**) Immunohistochemistry with anti-GROα; (**g**,**h**) Immunohistochemistry with anti-EDN1; (**i**,**j**) Immunohistochemistry with anti-EDNBR; (**k**,**l**) Immunohistochemistry with anti-SCF; (**m**,**n**) Immunohistochemistry with anti-c-KIT; (**o**,**p**) Immunohistochemistry with anti-IL-1α; (**q**,**r**) Immunohistochemistry with anti-TNF.

**Figure 2 ijms-20-03666-f002:**
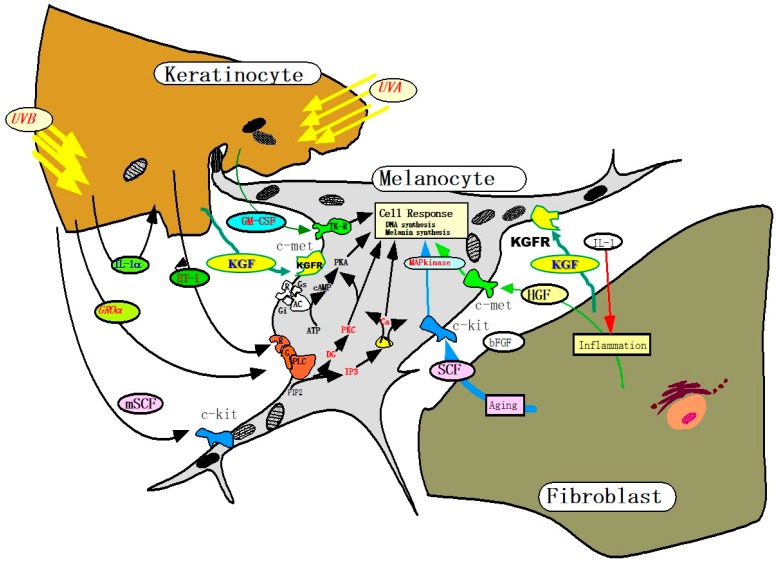
Melanogenic paracrine cytokine networks between skin cells [8,25,26,30].

**Figure 3 ijms-20-03666-f003:**
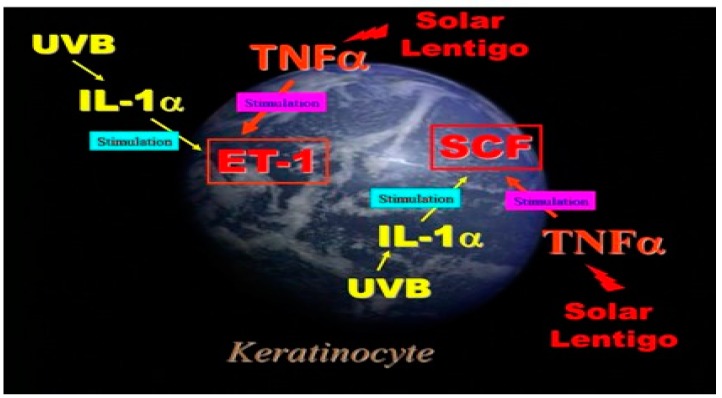
Autocrine cytokine linkages in UVB-exposed human keratinocytes and in the epidermis of SL.

**Figure 4 ijms-20-03666-f004:**
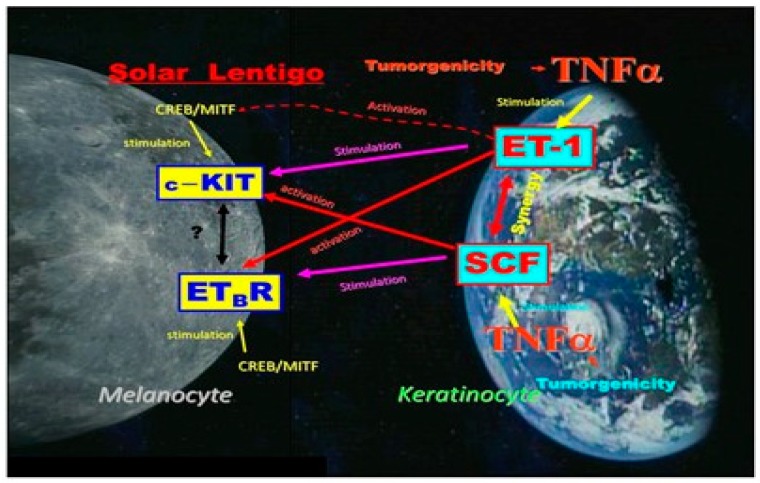
A summary of the complex mutual relationships between SCF and EDN1 linkages in the lesional epidermis of SLs.

**Figure 5 ijms-20-03666-f005:**
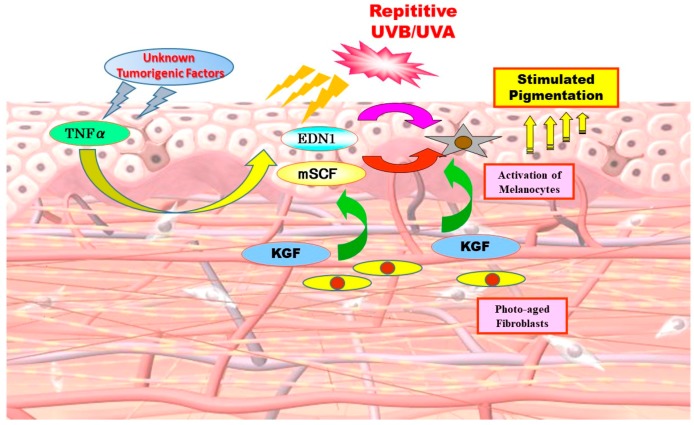
The biological sequence for the hyperpigmentation mechanism involved in SLs [2,4,8,25].

**Figure 6 ijms-20-03666-f006:**
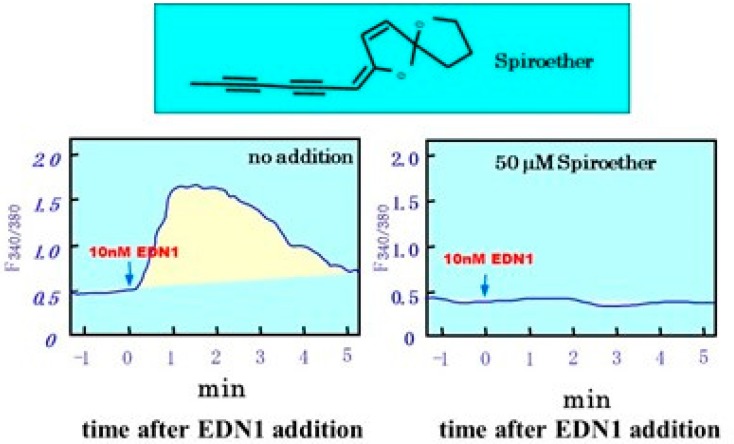
Inhibitory effects of spiroether on EDN1 induced intracellular calcium mobilization [1].

**Figure 7 ijms-20-03666-f007:**
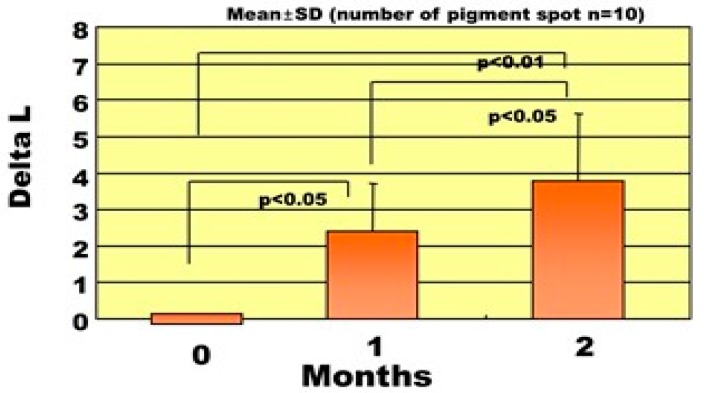
Changes in the ΔL values of SLs after treatment for two months [1].

**Figure 8 ijms-20-03666-f008:**
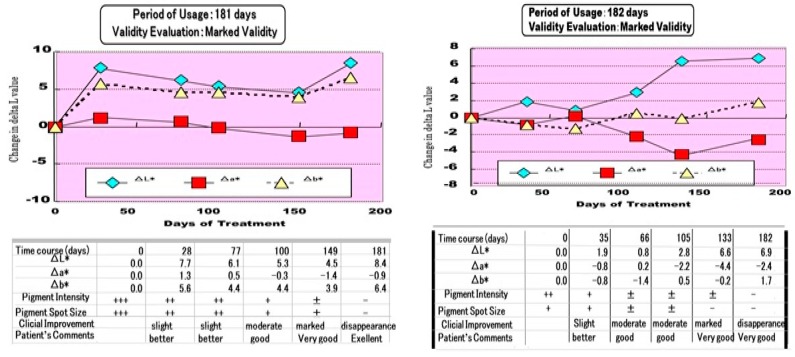
Two clinical cases where the SLs completely disappeared [1].

**Figure 9 ijms-20-03666-f009:**
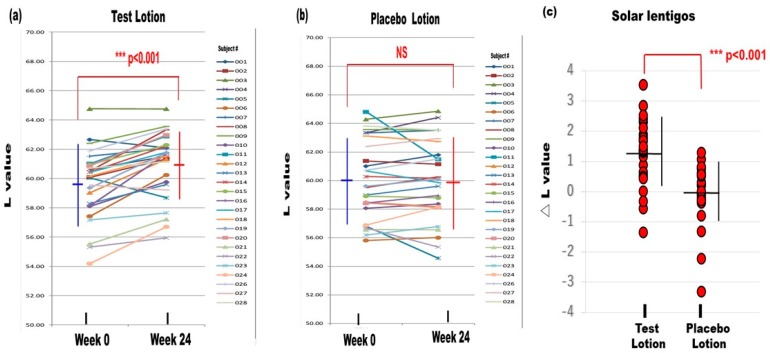
Changes in the L values of SLs after treatment for 24 weeks [45]. (**a**) Test lotion, *N* = 27, (**b**) Placebo lotion, *N* = 27, (**c**): △L values between weeks 0 and 24. *N* = 27, *** *p* < 0.001.

**Table 1 ijms-20-03666-t001:** Changes in the expression of cytokines, chemokines, and receptors in the lesional epidermis of several hyperpigmentary disorders when compared with non-lesional skin [8,25,30].

→Epidermis →Dermis	Cytokine/Chemokine								Receptor		
	ET-1	SCF	GROa	HGF	bFGF	KGF	IL-1a	TNFa	EDNBR	c-KIT	KGFR
**UVB melanosis**	↑	↑	→	→	↑	↑	↑	→	↑	→	↑
**Solar Lentigo**	↑	↑	→	?	→	↑	↓	↑	↑	↗	↑
**Seborrehoic Keratosis**	↑	⇡	?	?	?	?	↓	↑	?	?	?
**Dermato-fibroma**	→ →	↑ →	→ →	↑ →	→ →	?	?	?	?	↑	?
**Café-au-lait** **macules**	→ →	↑	?	↑	→	?	?	?	?	?	?
**Riehl’s Melanosis**	?	?	↑	?	?	?	↑	?	?	?	?
**Vitiligo Vulgaris**	↑	↑	?	→	→	?	?	?	→	↓	?

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
