# Peer review of "Melanocyte Activation Mechanisms and Rational Therapeutic Treatments of Solar Lentigos"

_ijms, 2019, doi:10.3390/ijms20153666_

Round 1

Reviewer 1 Report

This is a well-written review of the role of Endothelin and SCF in regulating pigmentation in solar lentigos. I have only a couple of comments.

In section 3.1 the authors state that because immuno-tyrosinase positive melanocytes were more numerous in SL areas of the skin, this proves that the number of melanocytes have increased in this area instead of just being "activated" in this area. This is an age-old argument. Are there more melanocytes or just more active melanocytes? I'm not sure that showing more immuno-tyrosinase positive melanocytes is proof that they are proliferating. Since the authors have shown that mRNA and protein levels for tyrosinase increase in SL regions, it seems possible that the increase in immunoreactivity is not due to increased proliferation but due to the increased expression of tyrosinase in existing melanocytes that makes them detectable by immunochemistry.

The authors focus on EDN as the hormone that is stimulating melanogenesis in SL melanocytes but do not talk about other mediators that could be stimulating these cells. Specifically, PGE-2 is an extremely potent stimulator of both melanocyte growth and melanogenesis, and could be involved in SL lesions. It would be useful for the authors to address the role of PGE-2 as well as other possibilities for melanocyte 'activators" in SL regions. Did some of their earlier published work look for other melanogenic stimulators in lesional skin? If so, it would be worthwhile in this review to talk about that. 

The authors cite mostly their own work, and very little from other researchers. For a review it would have been useful to have a broader citation base.

Author Response

Reviewer No1

This is a well-written review of the role of Endothelin and SCF in regulating pigmentation in solar lentigos. I have only a couple of comments.

In section 3.1 the authors state that because immuno-tyrosinase positive melanocytes were more numerous in SL areas of the skin, this proves that the number of melanocytes have increased in this area instead of just being "activated" in this area. This is an age-old argument. Are there more melanocytes or just more active melanocytes? I'm not sure that showing more immuno-tyrosinase positive melanocytes is proof that they are proliferating. Since the authors have shown that mRNA and protein levels for tyrosinase increase in SL regions, it seems possible that the increase in immunoreactivity is not due to increased proliferation but due to the increased expression of tyrosinase in existing melanocytes that makes them detectable by immunochemistry.

Response:

Thank you for your good comments as to whether the increased number of immune-positive melanocytes is due to proliferation or increased tyrosinase. I have checked this point by reading the related papers and rewritten the parts according to the data with references No 5, 6, 7 as marked by red in the revised manuscript with tracking remarks (page 4).

The authors focus on EDN as the hormone that is stimulating melanogenesis in SL melanocytes but do not talk about other mediators that could be stimulating these cells. Specifically, PGE-2 is an extremely potent stimulator of both melanocyte growth and melanogenesis, and could be involved in SL lesions. It would be useful for the authors to address the role of PGE-2 as well as other possibilities for melanocyte 'activators" in SL regions. Did some of their earlier published work look for other melanogenic stimulators in lesional skin? If so, it would be worthwhile in this review to talk about that.

The authors cite mostly their own work, and very little from other researchers. For a review it would have been useful to have a broader citation base.

Response:

I agreed with the reviewer that for a review it would have been useful to have a broader citation base. According to the reviewer’s comments, I have added many related studies by other researchers including PGE2 study as marked by red in references (No 5, 6, 7, 17, 18, 19, 23, 24, 28) and in revised manuscript with tracking remarks (mainly page 5 and 10).

Reviewer 2 Report

It is a review article that describe all findings that author did for SLs. It can be interesting but there are two big problems of this review article.

1st, author mainly focus on author’s study. Other studies performed by other researcher’ should be included and discussed together to show present situation of SL investigation.

2nd, author describe author’s work very well. But, it contain too much detail which are not compatible with review article. Then, it should be very shortened.

Finally, English is redundant. It can be shortened and more packed.

Abstract

Abstract should be like an abstract.

It is not necessary to say what technique is used in this study. In addition, author should consider it is review article.

Unknown tumorigenic factor? – it is not conceivable. Speculation without evidences need to move to discussion.

There are very details in abstract how M. Chammolilla was screened and what effect were obtained.

Abstract should be very shortened to deliver main message.

Introduction

Provide evidence that SL have cumulative DNA damage.

Introduction of this study can be more short and simpler. Clinical data of Teikyo University is not necessary because everybody know that SL is very common aging pigmentary disorders.

2. Clinical characteristics

Line 56: of skin is not necessary

Line 57: The histochemistry of SL lesions stained with: can be removed without any misunderstanding

Legend of Figure 1 is very redundant.

Line 70: against: English editing

Line 71: With Lever’s book; can be removed

Line 76: we hypothesized that KC trigger …: There is no enough evidence in this part to support this hypothesis.

Page 3.

3.2.

Line 80: we have elucidate that : is not necessary

Line 81 – 87 (Mainly, … of SLs). : is not necessary or it should be very shortened.

Figure 2 need to be more simpler to understand this article.

3.3.

This part comprised beginning description about UVB effects and what was observed in SL. Please it should be more compact. Too long.

Generally, figure is not good to see. Please edit it.

Table 1 needs to be more clear to see the contents.

Therapeutic topical approach

It should be more shorter and simple. It is a review article. But, contents for Chamomillar cover 4 page and content of APS cover 3 page. Any tyrosinase inhibitor can be effective, In addition, two many figures are included to show the effects of Chammomillar and APS. It is strongly recommended to show each representative one figure for each agent.

Author Response

Reviewer No2

It is a review article that describe all findings that author did for SLs. It can be interesting but there are two big problems of this review article.

1st, author mainly focus on author’s study. Other studies performed by other researcher’ should be included and discussed together to show present situation of SL investigation.

2nd, author describe author’s work very well. But, it contain too much detail which are not compatible with review article. Then, it should be very shortened.

Finally, English is redundant. It can be shortened and more packed.

Response:

According to the reviewer’s comments, I have added many related studies by other researchers as marked by red in references (No 5, 6, 7, 17, 18, 19, 23, 24, 28) and in revised manuscript with tracking remarks (mainly page 5 and 10).

In light of the reviewer’s suggestions, I have shortened our detailed data and English.

Abstract

Abstract should be like an abstract.

It is not necessary to say what technique is used in this study. In addition, author should consider it is review article.

Response:

In light of the reviewer’s comments, I have deleted technical parts in Abstract as revealed by red in abstract of the revised manuscript with tracking remark.

Unknown tumorigenic factor? – it is not conceivable. Speculation without evidences need to move to discussion.

Response: According to the reviewer’s critiques, I have deleted the related sentences in Abstract as revealed by red in abstract of the revised manuscript with tracking remark.

There are very details in abstract how M. Chammolilla was screened and what effect were obtained.

Abstract should be very shortened to deliver main message.

Response:

I have deleted some sentences related to what the reviewer pointed out as revealed by red in abstract in the revised manuscript with tracking remarks.

Introduction

Provide evidence that SL have cumulative DNA damage.

Introduction of this study can be more short and simpler. Clinical data of Teikyo University is not necessary because everybody know that SL is very common aging pigmentary disorders.

Response:

I have deleted some sentences related to what the reviewer pointed out in introduction as revealed by red in the revised manuscript with tracking remark.

2. Clinical characteristics

Line 56: of skin is not necessary

Line 57: The histochemistry of SL lesions stained with: can be removed without any misunderstanding

Legend of Figure 1 is very redundant.

Line 70: against: English editing

Line 71: With Lever’s book; can be removed

Line 76: we hypothesized that KC trigger …: There is no enough evidence in this part to support this hypothesis.

Response:

According to the reviewer’s comments and critiques, I have modified or deleted the related sentences as revealed by red in the corresponding pages in the revised manuscript with tracking remarks.

Page 3.

3.2.

Line 80: we have elucidate that : is not necessary

Line 81 – 87 (Mainly, … of SLs). : is not necessary or it should be very shortened.

Figure 2 need to be more simpler to understand this article.

3.3.

This part comprised beginning description about UVB effects and what was observed in SL. Please it should be more compact. Too long.

Generally, figure is not good to see. Please edit it.

Table 1 needs to be more clear to see the contents.

Response:

According to the reviewer’s suggestions, I have modified figure quality and several sentences as revealed by red in the revised manuscript with tracking remarks.

Therapeutic topical approach

It should be more shorter and simple. It is a review article. But, contents for Chamomillar cover 4 page and content of APS cover 3 page. Any tyrosinase inhibitor can be effective, In addition, two many figures are included to show the effects of Chammomillar and APS. It is strongly recommended to show each representative one figure for each agent.

Response:

According to the reviewer’s comments, l have deleted several related figures as revealed by red in the revised manuscript with tracking remarks.

Reviewer 3 Report

Having read the manuscript "Melanocyte activation mechanisms and rational therapeutic treatments of solar lentigos" by Imokawa, I have the following comments.

The author describes the effect cytokines and GF have on melanin synthesis in melanocytes, as well as in the pathogenesis of solar lentigo.  The last part of the review looks at plant extracts and derived chemicals on the treatment of patients with SL.

(1)  The reference list lacks papers from other authors which is a major failing of this manuscript.  There needs to be reference to other recent relevant papers to enhance the quality of the review article rather than a summary of an individual's research career.

(2)  Why are none of the figures in the review referenced, as they are taken from published articles.   The legends on most figures need to be expanded to fully describe what the figure represents.

(3) Table 1 is copied from another source and is blurry, in fact a number of images are blurry (eg. Figs 5, 6, 8, 9, 11) and should therefore be redrawn. Fig 4 looks like it has been taken from a Powerpoint slide, and is difficult to read, it needs to be redrawn.

(4) There are too many images in this review article, and some of these images can be combined and those that are taken from research papers should be referred to and the images not reproduced.

(5) The work on the  M. chamomilla extracts as well as spirometer in treating patients who have SL appear to be from studies you have undertaken and published earlier, and as such a brief summary of the outcomes is needed along with that from other studies on SL or skin whitening based on plant extracts.

Author Response

Reviewer No3

Having read the manuscript "Melanocyte activation mechanisms and rational therapeutic treatments of solar lentigos" by Imokawa, I have the following comments.

The author describes the effect cytokines and GF have on melanin synthesis in melanocytes, as well as in the pathogenesis of solar lentigo.  The last part of the review looks at plant extracts and derived chemicals on the treatment of patients with SL.

(1)  The reference list lacks papers from other authors which is a major failing of this manuscript.  There needs to be reference to other recent relevant papers to enhance the quality of the review article rather than a summary of an individual's research career.

Response:

According to the reviewer’s comments, I have added many related studies by other researchers as marked by red in references (No 5, 6, 7, 17, 18, 19, 23, 24, 28) and in revised manuscript (mainly page 5 and 10).

(2)  Why are none of the figures in the review referenced, as they are taken from published articles.   The legends on most figures need to be expanded to fully describe what the figure represents.

Response:

According to the reviewer’s comments, I have added references to all figures.

I think that fully described figure legends are not necessary. If readers are interested in some data, they should come to the original paper to know them in details.

(3) Table 1 is copied from another source and is blurry, in fact a number of images are blurry (eg. Figs 5, 6, 8, 9, 11) and should therefore be redrawn. Fig 4 looks like it has been taken from a Powerpoint slide, and is difficult to read, it needs to be redrawn.

Response:

According to the reviewer’s suggestions, I have deleted Fig 4 and modified figure quality in related figures as revealed by red in the revised manuscript.

(4) There are too many images in this review article, and some of these images can be combined and those that are taken from research papers should be referred to and the images not reproduced.

Response:

According to the reviewer’s comments, I have deleted totally 9 figures in the revised manuscript.

(5) The work on the M. chamomilla extracts as well as spirometer in treating patients who have SL appear to be from studies you have undertaken and published earlier, and as such a brief summary of the outcomes is needed along with that from other studies on SL or skin whitening based on plant extracts.

Response:

I understand the reviewer’s requests but in this review article, I am not able to fulfill the requirements.

Round 2

Reviewer 2 Report

This manucript need more fine edting and correction.

Reviewer 3 Report

The author has revised the manuscript and inserted other references which have improved the  quality of the manuscript. I thank the author for making these changes.